# MixNN: A Design for Protecting Deep Learning Models

**DOI:** 10.3390/s22218254

**Published:** 2022-10-28

**Authors:** Chao Liu, Hao Chen, Yusen Wu, Rui Jin

**Affiliations:** 1Department of Computer Science and Electrical Engineering, University of Maryland, Baltimore, MD 21250, USA; 2Department of Computer Science, University of Miami, Coral Gables, FL 33146, USA

**Keywords:** deep learning, distributed system, privacy, mix network

## Abstract

In this paper, we propose a novel design, called MixNN, for protecting deep learning model structure and parameters since the model consists of several layers and each layer contains its own structure and parameters. The layers in a deep learning model of MixNN are fully decentralized. It hides communication address, layer parameters and operations, and forward as well as backward message flows among non-adjacent layers using the ideas from mix networks. MixNN has the following advantages: (i) an adversary cannot fully control all layers of a model, including the structure and parameters; (ii) even some layers may collude but they cannot tamper with other honest layers; (iii) model privacy is preserved in the training phase. We provide detailed descriptions for deployment. In one classification experiment, we compared a neural network deployed in a virtual machine with the same one using the MixNN design on the AWS EC2. The result shows that our MixNN retains less than 0.001 difference in terms of classification accuracy, while the whole running time of MixNN is about 7.5 times slower than the one running on a single virtual machine.

## 1. Introduction

Privacy protection of deep learning (DL) models is important for guarding commercial and intellectual property; for example, a financial company may hold a private model which can facilitate stock investment; leakage of such a model causes huge loss [1]. Protecting DL models contains both model structure and model parameters.

Privacy concerns of a DL model occur when deploying a model locally or on a cloud server. Deploying DL models in a local machine very probably leaks all the details of models to hackers and malicious colleagues. Meanwhile, machine learning as a service (MLaaS), such as, Amazon Machine Learning services [2], Microsoft Azure Machine Learning [3], Google AI Platform [4], and IBM Watson Machine Learning [5], enables customers to use powerful DL tools and computation resources by deploying their DL models on the cloud. Even though private data from a client can be protected by using Intel SGX [6] or homomorphic encryption (HE) [7], an obvious problem is that the model may still be opened to some providers. Another issue is that the malicious cloud controller can easily steal the DL model by checking the codes, or can obtain a close model by generating querying results and searching the DL model space [8,9]. In 2016, Tramer et al. [9] proposed the model extraction attack. Attackers can use the “shadow training” method to train a new model with the same functionality as the original model. In 2015, Fredrikson et al. [10] proposed the estimation attack scheme. Attackers can estimate whether certain characteristic values $x_{i}$ belong to certain training data sets. For example, in some applications that provide information concerning users’ locations, attackers can learn about users’ private geographic location and other information. Shokri et al. [11] proposed a member extraction attack. Attackers can use black-box attack to identify whether the target model prediction is on the training set or non-training set. There are many other attacks, such as model memorization attack, inference attack, feature estimation attack. It is a challenge to protect the privacy of deep learning model.

Based on the above issues, we propose the MixNN to explain *how to deploy a DL model on a powerful AI platform while protecting model privacy.*

Inspired by split learning [12], MixNN distributes each layer in a DL model on one server. We assume that an adversary cannot control most of the layers. In this way, it prevents an adversary from acquiring the whole model structure and model parameters. We will discuss a special scenario where an adversary controls both the first layer and the last layer in MixNN in Section 5.1. We consider MixNN only on a cascade topology where forward and backward propagation must proceed through consecutive layers.

One problem in this design is that one server controlled by an adversary in this cascade can figure out who the other servers are via decoding the message flow; then, this malicious server could tamper with other honest servers by rewarding them (e.g., bitcoin) for exchanging information. Another situation is that mutually acknowledged servers can cooperate together to disclose the sensitive model and data for common interest. If we do not have a way to hide the detailed physical address among these servers, an adversary can easily acquire the information by passively listening to the channel among these mutually acknowledged servers. Thus, the model structure and parameters of these layers on these malicious servers are exposed.

To tackle this issue, we adapt a method from mix networks proposed by Chaum [13]. In mix networks [14], each message is encrypted to each mix node using public key cryptography. The resulting ciphertext is layered like an onion. Each mix node strips off its own layer of encryption to reveal where to send the message next. We use a layer to denote one mix node in MixNN. In Figure 1, we take a communication process from layer 1 to layer *n* as an example, layer 2 only knows that it receives messages from layer 1. It then uses its secret key to decrypt the message flow from layer 1 and obtains layer 3’s physical address, and finally it sends the message to layer 3. Layer 2 has no knowledge about the address from layer 4 to layer *n*, assuming no failures occur. In MixNN, we use this approach to pack the message on the client slide to hide the detailed communication process among non-adjacent layers. In summary, even some layers are controlled by an adversary but it is hard for it to locate other honest layers in the network.

Training a DL model, however, is different from using mix networks to realize anonymous communication. The *i*th layer in the model should compute the input set $Z_{i}$ for the next layer in the forward propagation, and update the parameter set $W_{i}$ in this layer by gradient descent during the backward propagation. In MixNN, layer *i* decrypts the $Z_{i-1}$ from layer i−1 to cause DL operations to proceed. It also decrypts the intermediate gradient $\frac{\partial l}{\partial z_{i}}$, where *l* represents the training loss and $z_{i}\in Z_{i}$, from layer i+1 to update $W_{i}$, and prepares the intermediate gradient $\frac{\partial l}{\partial z_{i-1}}$ for layer i−1. We use the chain rule in updating parameters and preparing gradients. Finally, layer *i* uses the next layer’s public key to encrypt forward or backward propagation messages. Our MixNN fits a cascade topology which is one of the structures [15] (cascade topology and free-routing topology) in mix networks. An adversary who is listening to the channel cannot learn both the DL computation result in layer *i* and any information of DL parameters and operations in other non-adjacent layers.

In summary, MixNN not only distributes layers in a DL model on different servers so that an adversary can hardly control the whole model structure and parameters, but also hides the communication address, layer parameters and operations, and forward and backward message flow among non-adjacent layers. Figure 1 shows the MixNN’s design. An adversary can control the server and passively listen to the channel between two servers. The square with color means a layer in a DL model and this colored layer is deployed on one server. All layers are constructed for one DL model. Some servers which do not have a layer on them are dummy servers; we will explain this in Section 2.2.

### 1.1. Related Work

Existing works [16,17,18,19,20] for protecting model privacy have tried to avoid leaking the model information to users by isolating users on the client side and the model on the server side. They also provide different strategies to secure model prediction results between the server and the client. Studies [20] and DeepSecure [17] protect model privacy by sending encrypted results back to the client. However, these methods either modify neural networks by replacing activation functions with polynomial approximations, or transform an original neural network to a Boolean circuit. Thus, the complexity concerning implementation is increased and the performance of the model might be discounted. Moreover, they take no account of the situation when the server itself is an adversary who could steal the whole model on the server side. Dowlin et al. proposed CryptoNets [18]. CryptoNets can be applied to encrypted data by using homomorphic encryption (HE). However, HE can be only used on arithmetic operations, such as multiplication and addition. Even though the author used a square function instead of a sigmoid function, the inference accuracy would be influenced. Liu et al. [21] introduced the MiniONN by using an oblivious neural network. This work can transform an existing model to an oblivious one, supporting privacy-preserving predictions. This protocol has two phases, an offline precomputation phase and an online prediction phase. However, the user computation overhead in the precomputation phase is linearly correlated with the number of neurons of the neural network model. Mo et al. [22] have suggested a framework that uses an edge device’s Trusted Execution Environment (TEE) in conjunction with model partitioning to limit the attack surface against DNNs. Shokri et al. [23] proposed privacy-preserving deep learning by sharing partial parameters during the training process. They focus on training deep networks, emphasize the importance of privacy, and address communication costs by only sharing a subset of the parameters during each round of communication; however, they also do not consider unbalanced and non-identity data, and the empirical evaluation is limited. The authors of [24] use additively homomorphic encryption for model parameter aggregation to provide security against the central server. A study [25] used the SMC framework for training machine-learning models with two servers and semi-honest assumptions. Ma et al. [19] discuss a similar scenario but split the neural network into two shares and place them in two servers. They use both HE and secure two-party computation protocols between two servers to preserve model privacy against attack on the server side. Ma et al. [19] and Rouhani et al. [17] focus only on inference of a pre-trained model. Compared to these previous works, our proposed design instead naturally protects model privacy from attack on the server side by decentralizing layers into different servers and hiding the server’s physical position. It does not require any modification or transformation of the neural network and is naturally applicable to the training phase.

Other current studies on model privacy protection commonly allow model users on the client side to keep certain layers of neural networks while servers keep the rest [12,26,27]. Although their primary target is to ensure data privacy on the client side, model privacy protection is implicitly involved as the adversary on the server only controls parts of the neural network. Nevertheless, model privacy is still at risk because the adversary can infer the other part of the model by analyzing message flows or occupying them. The split learning [12] mentioned the multi-hop configuration, which is similar to our decentralized deployment. However, they neither give detailed methods to deploy such a DL model nor focus on model privacy. Our proposed MixNN deploys layers in different servers and uses the method from mix networks to prevent the adversary from controlling the whole model structure and model parameters.

### 1.2. Contributions

MixNN is a novel design for protecting model structure and parameters. Compared with previous works, MixNN decentralizes layers in a DL model on different servers instead of two parties (some layers on the client side and the rest on the server side). This distributed method decreases the possibility that an adversary will control the whole structure and parameters of a model.MixNN is the first design to use the ideas from mix networks for hiding real “identities” of non-adjacent layers in a cascade topology in DL structures. In this design, MixNN actually isolates every layer in a black box. An adversary can hold some black boxes and obtain parameters and operations but they cannot locate and control all of them. When transferring a message layer by layer, each layer encrypts forward and backward propagation messages to avoid leaking model information to the adversary who is passively listening to the channel.We provide a detailed description for deploying MixNN. It explains how to decentralize layers and how to use the method from mix networks to pack a message in different DL phases. There are four phases in MixNN: model initialization, forward propagation phase, backward propagation phase, and testing phase, separately. The implementation follows the description of MixNN. Compared with the same neural network deployed in a single server on AWS EC2, we show that the MixNN has less than 0.001 difference in terms of classification accuracy, while the whole running time is about 7.5 times slower than the one run in a single virtual machine.

## 2. MixNN Design

### 2.1. Adversary Model

There are two parties in the MixNN, namely the designer and servers. A designer is the one who deploys the DL model and processes his or her private data. The servers hold the model layers and provide computation tools and resources. Layers are distributed on different servers and all layers are constructed as a DL model. We consider an adversary who can control a subset of *n* layers in the system and its goal is to simulate a model $f^{\prime }\left(x\right)$ which is approximately the same as the initial function f(x). We assume each pair of servers is connected by an authenticated point-to-point channel. An adversary who can launch denial-of-service (DOS) attacks is not included in this paper. We also assume that a designer who deploys his/her own DL model using his/her own private data is honest.

### 2.2. Setup

There are *m* servers running in a pool. The designer can acquire servers’ information, such as location, configuration, communication speed, price per hour, and so on. The designer can randomly select *n* servers for deploying the DL model from *m* (m≫n). Among the *n* servers, *p* of them contain actual layers (servers) who perform DL operations and *r* are dummy servers, namely n=p+r. A way to choose these *p* servers could be based on servers’ historical logs, for example, their crash history and performance. He/she assigns *p* actual layers in *p* servers. Remaining dummy servers could perform “obscure” operations, for example, transferring messages, or passing through the activation function, e.g., Rectified Linear Unit (ReLU). The *r* dummy layers can be randomly distributed among these *p* actual layers. Adding dummy layers among actual layers decreases the possibility of an adversary controlling the actual layers and acquiring information for simulating the same DL model.

*m* servers should register to an authority (this authority could be distributed). Every server in the pool generates its own key pair $pk_{i}/sk_{i}$ where $pk_{i}$ is its public key and $sk_{i}$ is its secret key, and the server publishes its $pk_{i}$ and keeps its $sk_{i}$ secretly. The *i*th server owns its unique $A_{i}$ address in the system and $A_{u}$ stands for a designer’s address.

The designer connects *n* servers as a fixed cascade. Only the designer in the system can pack an IP address in the message. To achieve this, we let the designer send a loop message to itself so that no one in the system can know all physical positions of the *n* servers (layers).

Besides distributing *r* dummy servers among *p* actual servers, we use a similar method called “loop message”, proposed in [28], in which a server (layer) in the system can send a dummy message to another server (layer) in the system. In this way, an adversary cannot know where the message comes from and what it is for.

### 2.3. Training Phase

The training phase contains three phases: model initialization phase, forward propagation phase, and backward propagation phase. The model initialization is executed only once at the beginning of the training. We set the training with multiple epochs and each epoch includes several iterations. In every iteration, MixNN causes one forward propagation and one backward propagation.

In MixNN, we set every package with the same length so that an adversary cannot tell which type of package it is. A package includes four segments, ($op,en_{m},en_{IP},padding$). op denotes which type of operation a layer carries out. It contains four types of operations: (1) op=0 means that the designer initializes every layer in a DL model and every layer has to build its corresponding part of the model; (2) op=1 stands for a forward propagation message; (3) op=2 denotes a backward propagation message; and (4) op=3 indicates a testing operation. $en_{m}$ means an encrypted message. $en_{IP}$ denotes an encrypted IP address, and adding padding segments is used to keep the package in a consistent length.

In Figure 2a, when a cascade is constructed, only adjacent layers know the previous and next layers’ IP address but they have no knowledge about other layers’ location. For example, layer 2 only knows layer 1 and layer 3’s physical IP address and their public keys, but it is hard for layer 2 to acquire layer *i*’s location since layer *i*’s location is wrapped in an inner part of the package. Layer *i*’s location can be acquired in layer i−1 assuming there are no failures. For simplicity, we only show the scenario, n=p; that is, the DL model does not include dummy layers in it.

#### 2.3.1. Model Initialization Phase

In the model initialization phase, the designer needs to distribute the DL model to *p* servers. For the layers which conduct DL operations, the designer wraps the operation type, size of parameters and IP address in a message. The designer packs the message as below and sends *c* to the first layer.
(1)$$\begin{array}{c} c=(c_{1},\;A_{1}),\\ c_{1}=E_{pk_{1}}\left(op=0,\;para_{1},\;c_{2},\;A_{2}\right),\\ c_{2}=E_{pk_{2}}\left(op=0,\;para_{2},\;c_{i-1},\;A_{i-1}\right),\\ \cdots \\ c_{i-1}=E_{pk_{i-1}}\left(op=0,\;para_{i-1},\;c_{i},\;A_{i}\right),\\ c_{i}=E_{pk_{i}}\left(op=0,\;para_{i},\;c_{n},\;A_{n}\right),\\ \cdots \\ c_{n}=E_{pk_{n}}\left(op=0,\;para_{n}\right). \end{array}$$

The $E_{pk_{i}}\left(data\right)$ means that the encryption algorithm *E* uses public key $pk_{i}$ to encrypt data and the ciphertext can only be decrypted by the corresponding secret key $sk_{i}$ with decryption algorithm *D* ($D_{sk_{i}}\left(E_{pk_{i}}\left(data\right)\right)=data$). $para_{i}$ is the size of the parameters, and $A_{i}$ is *i*th layer’s address. We can also use the signcryption scheme [29,30].

After packing the message above, the designer sends *c* to layer 1 according to layer 1’s IP address $A_{1}$. Layer 1 can use its secret key to decrypt the message received from the designer, obtain the operation type op=0, the size of parameter $para_{1}$, a ciphertext $c_{2}$, and layer 2’s address $A_{2}$. Layer 1 builds its corresponding part of the model and optimizer and sets its own input size of parameter with $para_{1}$. The parameters of the optimizer, such as learning rate and momentum, are the same for all actual layers. We can also send these parameters, but it is unnecessary here. Layer 2 to layer *n* operate the same model initialization as layer 1. After finishing the model initialization phase, each layer in a DL model is like what is shown in Figure 2b.

#### 2.3.2. Forward Propagation Phase

The forward propagation phase is similar to the initialization phase. However, both forward propagation and backward propagation should be iteratively executed within multiple epochs. Before transmitting the message to the next layer, layer *i* needs to compute $Z_{i}$, encrypt it with the next layer’s public key, and pack it with $c_{i}$. The designer packs the forward propagation message as below and sends *c* to the first layer.
(2)$$\begin{array}{c} c=(E_{pk_{1}}\left(data\right),\;c_{1},\;A_{1}),\\ c_{1}=E_{pk_{1}}\left(op=1,\;c_{2},\;A_{2}\right),\\ c_{2}=E_{pk_{2}}\left(op=1,\;c_{i-1},\;A_{i-1}\right),\\ \cdots \\ c_{i-1}=E_{pk_{i-1}}\left(op=1,\;c_{i},\;A_{i}\right),\\ c_{i}=E_{pk_{i}}\left(op=1,\;c_{n},\;A_{n}\right),\\ \cdots \\ c_{n}=E_{pk_{n}}\left(op=1,\;supervised\;signals,\;A_{u}\right). \end{array}$$

Different from other phases in the training, we can see that the designer should pack the data as well as the supervised signal, and send the package to layer 1. The data privacy is not the core part of our paper but we discuss some methods to protect data privacy in Section 5.

When layer 1 receives *c* from the designer, it decrypts the ciphertext and obtains the data, operation type op=1, ciphertext $c_{1}$, and address $A_{2}$. Layer 1 inputs data to the DL operation in this layer to compute the result $Z_{1}$. Then, layer 1 uses layer 2’s public key $pk_{2}$ to encrypt $Z_{1}$, packs it with $c_{1}$, and sends ($E_{pk_{2}}\left(Z_{1}\right),c_{2}$) to layer 2. Layer 2 to layer n−1 repeat the same steps as what layer 1 does; for example, after decrypting the ciphertext from layer i−2, layer i−1 computes $Z_{i-1}$ and sends ciphertext ($E_{pk_{i}}\left(Z_{i-1}\right),c_{i}$) to layer *i*.

Layer *n* calculates the training loss using $Z_{n-1}$ and the supervised signal, then encrypts the loss *l* and sends it back to the designer. The supervised signal is visible only at the last actual layer as it is the most inner part of the package. The designer can also hold the loss layer and supervised signals by him or herself, and hence supervised signals are protected if required. The forward propagation phase is shown in Figure 2c.

#### 2.3.3. Backward Propagation Phase

The backward propagation instead starts from layer *n* to layer 1, which is different from the above two phases. We here only consider gradient descent-based methods in updating the DL model parameter. Layer *i* receives the intermediate gradient $\frac{\partial l}{\partial z_{i}}$ computed in layer i+1, calculates the intermediate gradient $\frac{\partial l}{\partial z_{i-1}}$ for the next layer i−1 using chain rule $\frac{\partial l}{\partial z_{i}}\left(\frac{\partial z_{i}}{\partial z_{i-1}}\right)$, and sends it to the next layer. The parameters $W_{i}$ in layer *i* are updated by first applying chain rule $\frac{\partial l}{\partial z_{i}}\left(\frac{\partial z_{i}}{\partial w_{i}}\right)$, and then performing gradient descent related operations. In this phase, the designer only packs the IP address for communication as below. MixNN does not pack any other information (e.g., data or supervised signals) in this phase, which is different from the other phases. Finally, the designer sends *c* to layer *n*.
(3)$$\begin{array}{c} c=(c_{n},\;A_{n}),\\ c_{n}=E_{pk_{n}}\left(op=2,\;c_{i},\;A_{i-1}\right),\\ \cdots \\ c_{i}=E_{pk_{i}}\left(op=2,\;c_{i-1},\;A_{i-1}\right),\\ c_{i-1}=E_{pk_{i-1}}\left(op=2,\;c_{2},\;A_{2}\right),\\ \cdots \\ c_{2}=E_{pk_{2}}\left(op=2,\;c_{1},\;A_{1}\right),\\ c_{1}=E_{pk_{1}}\left(op=2,\;A_{u}\right). \end{array}$$

Layer *i* receives a message from layer i+1; it then decrypts the ciphertext and obtains the operation type op=2, intermediate gradient $\frac{\partial l}{\partial z_{i}}$, ciphertext $c_{i}$, and address $A_{i-1}$. Layer *i* calculates $\frac{\partial l}{\partial z_{i-1}}$ and encrypts it with $pk_{i-1}$. Finally, layer *i* packs ($E_{pk_{i-1}}\left(\frac{\partial l}{\partial z_{i-1}}\right),c_{i-1}$) and sends it to layer i−1. The backward propagation is shown in Figure 2d.

### 2.4. Testing Phase

When the training phase is finished, the designer can perform testing or inference using his or her own metric. The procedure is pretty similar to the forward propagation phase while an input to the metric is needed other than a loss from the model. The packing message is shown below. The testing is only a one-way process, the designer sets the operation type with op=3 and decides on an ending layer to generate the corresponding input. After one forward propagation, MixNN sends it back to the designer.
(4)$$\begin{array}{c} c=(E_{pk_{1}}\left(data\right),\;c_{1},\;A_{1}),\\ c_{1}=E_{pk_{1}}\left(op=3,\;c_{2},\;A_{2}\right),\\ c_{2}=E_{pk_{2}}\left(op=3,\;c_{i-1},\;A_{i-1}\right),\\ \cdots \\ c_{i-1}=E_{pk_{i-1}}\left(op=3,\;c_{i},\;A_{i}\right),\\ c_{i}=E_{pk_{i}}\left(op=3,\;c_{n},\;A_{n}\right),\\ \cdots \\ c_{n}=E_{pk_{n}}\left(op=3,\;A_{u}\right). \end{array}$$

Take the classification task using probability as an example; the designer can let layer n−1 send the output of the softmax function back to the client and use it to judge the classification performance using various metrics such as the confusion matrix, precision, recall and F1 score.

## 3. Evaluation

### 3.1. Experiment Settings

We compare the performance and efficiency of a neural network with the same one using MixNN design in the MNIST handwritten digits classification task. In total, 30k training digits and 10k test digits are used for training and inference. The performance is defined as the classification accuracy, which is the proportion of correct predictions of the test dataset. The efficiency is measured using running time.

We employ a multilayer perceptron (MLP) with each layer’s configuration as listed in Table 1. It is trained using negative log-likelihood (NLL) loss in server (layer) 5 with the logarithm of probabilities (LogSoftmax) from server (layer) 4. We intentionally set servers (layers) 4 and 5 with no parameter to simulate a more flexible situation, as our MixNN allows designers to further split or merge operations in different layers of a DL model. The optimization method is the stochastic gradient descent (SGD) with mini-batch size 64, learning rate 0.01 and momentum 0.9. We use Pytorch to implement these settings.

The entire MixNN library is written using Python language. We use Python pycrypto as our crypto library and the public key encryption scheme is RSA with 2048 key length. We deploy MixNN on Amazon AWS. Each instance is run in Ubuntu 16.04 version 43.0 for deep learning. We use t2.micro with one vCPU and 1 GB memory. We run all instances in the same region (Virginia).

### 3.2. Results and Analysis

We show the classification accuracy and running time of training with different epochs. We name MLP and MixNN for two different settings in the results for simplicity. In Figure 3a, we can observe that the differences between classification accuracy of MixNN and MLP are always less than 0.001 in each epoch; thus, our MixNN keeps almost the same performance in MLP in this task. The reason is obvious, as MixNN does not modify the MLP during the training or inference, and we only have different parameter initialization and data shuffling in two settings.

In Figure 3b, we can see that the running time of MixNN of each epoch is always higher than its counterparts in MLP, and it is 7.5 times higher than the MLP case on average. The reason is that MixNN spends more time on transmitting messages between layers (servers) as well as encrypting and decrypting message flow, and the designer side needs to pack the messages twice in an iteration.

## 4. Security Analysis

We explain how MixNN resists the following attacks.

### 4.1. Crash Failure

A layer on a server may crash. This degrades the performance of MixNN, especially when the crash occurs in the training phase. We use the following method to defend against this attack. We define *t* as the maximum communication time when transferring a message between two servers, and δ as the average time at which a server proceeds with a message. We denote *n* as the total number of servers (layers).

A designer sets a time bound *T* (T>>nδ+(n−1)t) when he or she sends the message to the first server or the last server.If the designer does not receive the response within time *T*, the designer realizes that a crash failure occurs.The designer cannot locate crashed servers. A simple way is that the designer replaces all servers in MixNN with other *n* servers.

Another method is that when a server in the cascade does not receive the response from its adjacent server, the server can report the failure to the designer. There are two scenarios here: (1) an honest server reports this failure; (2) a malicious server reports this to achieve its goal such as decreasing the credits of an honest server. In MixNN, the designer cannot distinguish between the two scenarios, and the simplest way is to replace both the servers. MixNN can use the same approach proposed by Hemi et al. [31] to isolate malicious servers before a cascade transfers the real message.

### 4.2. Byzantine Failure

An adversary dominates a server in a cascade. It can acquire one layer’s structure and parameters in a model. Besides that, the adversary can also modify, add or delete the real message [32] which should be transferred to other layers. Thus, the correctness of a model is affected.

In MixNN, a designer cannot verify each layer’s input and output for locating the faulty layer. Meanwhile, verifying each layer’s result in every iteration needs more time. In order to guarantee the correctness of the model, we use a simple method whereby a designer verifies the performance of the model in the testing phase. If he/she finds any problems in that phase, the designer should replace the current *n* layers with new ones.

In our future work, we will consider whether non-interactive zero knowledge proof [33,34] can be used to verify each layer’s input and output.

### 4.3. Model Privacy

**Theorem** **1.**
*The MixNN satisfies the security definition of model privacy.*


**Proof.** The definition of model privacy requires that adversary A cannot simulate a model $f^{\prime }\left(x\right)$ which is approximately the same as the initial model f(x). In the adversary model, we have two assumptions: (1) the designer side is honest; (2) an adversary A controls most of the layers in a DL model.For assumption 1, we assume that an adversary A cannot acquire the private data the way that a designer configures layers in a DL model, the number of layers in a DL model, and the construction of cascade (the detailed physical address of these layers) on the designer side. For assumption 2, we assume that an adversary A cannot control most of layers in a DL model and adversary A cannot control both the first layer and the last layer. There is no restriction with regard to how many layers are faulty, for example, 1/3 or 1/2 of total layers. Under these assumptions, we prove that our design satisfies the model privacy in the training phase. We do not consider the model privacy in the testing phase where an adversary can query the model.*We first consider that an adversary A controls one layer i (i∈1,⋯,n) in a DL model.* We assume that operations with parameters are in $\sigma (W_{i}z_{i-1}+b_{i})$ format among all layers, where σ represents the nonlinearity. In the training phase, adversary A can acquire the input $Z_{i-1}$ and output $Z_{i}$ of layer *i*, and intermediate gradient $\frac{\partial l}{\partial z_{i}}$, where *l* represents the training loss and $z_{i}\in Z_{i}$, and intermediate gradient $\frac{\partial l}{\partial z_{i-1}}$. Then, the A is able to discover the $W_{i}$ and $b_{i}$, the number of rows in $W_{i-1}$, the dimension of $b_{i-1}$, and the number of columns in $W_{i+1}$. Even though adversary A can acquire the above information from layer *i*, it is hard for him/her to infer the other layers’ structures and parameters with our design. When the number of layers increases, the probability of adversary A simulating an $f^{\prime }\left(x\right)$ is negligible.*We then focus on the situation in which f layers are occupied by the adversary A.* We again assume that all layers with parameters have the same type of operations mentioned above. There are two cases below.**Case (i)** The adversary A does not know the position of layers in a cascade.Apparently, to adversary A, *f* layers are distributed randomly in this case. A indeed knows the parameters in *f* layers and their adjacent layers’ parameter dimensions. However, adversary A cannot figure out what these layers are and how to combine and construct them as a DL model. When there are more layers in a DL model, knowing these *f* layers is not very helpful for adversary A with regards to simulating an $f^{\prime }\left(x\right)$.**Case (ii)** Adversary A knows the position of layers in a cascade.The most severe attack in this case is shown in Figure 4. Adversary A knows the detailed position of a cascade which constructs a DL model in the model initialization phase and successfully dominates layers 2, 4 and 6. In the training phase, adversary A knows not only the structures and parameters of 2, 4 and 6 but also the size of the parameters in layers 3 and 5 and the input as well as the output of these two layers. Therefore, adversary A can find parameters in these two layers by model extraction methods [8,9]. This means that adversary A knows n−2 layers between the first layer and the last layer.However, the model privacy is kept by the first layer and the last layer. The raw data and loss are preserved secretly; hence, adversary A cannot obtain them or use them to simulate an $f^{\prime }\left(x\right)$. Without the loss layer, the adversary cannot know what this model is for.This completes the proof of Theorem 1. □

## 5. Discussion

### 5.1. Model Privacy

Although the MixNN decentralizes and hides model related information so that an adversary cannot fully obtain them for simulating an approximate DL model, a more serious case is that the adversary can successfully occupy the first and last layer, which perform DL operations in the current structure. As the input to the model on the first layer and loss as well as supervised signals on the last layer are exposed, the adversary can easily simulate an approximate model using this information.

In order to avoid this case, the designer can keep one of these two layers or both of them on his or her own hands. Therefore, the adversary cannot fully acquire the input, loss or supervised signals.

### 5.2. Data Privacy

Federated learning [35] and split learning [12] facilitate distributed collaborative learning without disclosing original training data. In MixNN, the designer should input the data to the first layer in the forward propagation phase and testing phase. If the first layer is unfortunately controlled by an adversary, the data can be accessed by that adversary. Therefore, the data privacy is not preserved. In order to solve this issue, we provide the following methods.

*Distribute the first layer on the client side*. This method is similar to the work mentioned in [36], and it avoids the data being leaked to an adversary. However, when the training is done, how to let other clients use this model is a problem since the first layer is on the designer side. If other clients want to test their dataset, they still need to transmit their data to the designer. A way to solve this is to use obfuscation [37] to obfuscate the first layer. After the training of a DL model is done, the designer can upload the obfuscated layer to his/her private cloud. Only authorized clients can access it, download it to his/her local machine, and use this part of the code as the entrance to the model.

*Trusted Execution Environments (TEEs)*. Another technique to protect confidentiality is using a TEE such as Intel SGX [38] or ARM TrustZone [39]. For example, SGX helps to increase protections for sensitive data even when an attacker has full control of the platform. The designer can send encrypted data into the SGX enclave and only the enclave can decrypt the data. However, accelerators such as GPUs do not support TEE, and the SGX has a limited memory size.

*Homomorphic encryption scheme*. Fully Homomorphic Encryption (FHE) [40] is a new class of encryption scheme that allows computing on encrypted data without decryption. FHE has been shown to be useful in many privacy-preserving applications such as image classification [18,41]. In the MixNN, we can use the same methods as above but change operations in each layer for processing the encrypted data. However, FHE suffers from two main problems: (1) high computational overhead; (2) limited arithmetic set (only addition and multiplication on encrypted data are naturally supported).

### 5.3. Efficiency Analysis

Compared with other works, such as [18,19,21,22], our design does not modify any original deep learning structure. Therefore, our design has almost the same accuracy compared with the original framework. However, as in study [18], which uses a homomorphic encryption scheme, it can only be used on arithmetic operations, and thus its inference accuracy is influenced. Study [21] transforms a model to an oblivious one and it has to divide the model into two phases, precompuation phase and online phase. The advantage of our work is in locating each layer in a black-box on different servers so that, if more server providers are honest, the privacy is preserved.

However, the latency in terms of training a model is high in our work and we analyze the issues and how to improve this in Section 5.4. We do not implement other DL models, such as VGG-16, but they can be established on our current implementation. For example, we have six layers in the current MLP implementation; that is to say, we use six servers to hold those six layers. For VGG-16, we can simulate 2 or 3 layers on one server.

### 5.4. Improvement for the Design of MixNN

#### 5.4.1. Another Configuration with MixNN Design

We only set one layer on one server in the current setting. For DL models with many more layers than the case in the experiment, we do not want too much degradation on the running time. One configuration is that we can randomly compose some adjacent layers on one server. Our next step is to test VGG 16 [42] with this configuration.

#### 5.4.2. Implementation

In the current implementation, every layer serves as both a server and a client. As a server, this layer is bound with an IP address and a port number, and it is listening to this channel via (IP, port) and waiting for the information. We do not use the multi-threading method to implement it. Hence, when the communication is frequent and the request buffer is full to this layer, it needs to wait for previous requests to be processed, and then it can settle other messages. This is another reason why running time becomes longer.

## 6. Conclusions

MixNN is a novel design for protecting model privacy. It divides a whole DL model into several parts; every part has its private part of models and parameters. This part in MixNN is a layer in a DL model, and all layers are fully decentralized and constructed as a cascade. Besides the structure above, MixNN utilizes the method from the mix network to hide the detailed communication address. Meanwhile, layer parameters and operations, and forward as well as backward message flow among a non-adjacent layer are also hidden. In the experiment, we compare MixNN with MLP and show that MixNN retains almost the same performance in terms of classification accuracy. Because MixNN spends time on transferring messages between servers (layers), encrypting and decrypting messages with a large size, and packing the messages into one iteration with two times, the running time of MixNN becomes larger. MixNN considers protection of the privacy of a deep learning model in the training phase. However, we do not provide the techniques to defend against the attacks in the inference phase, such as the model extraction attack. We also do not consider how to allow the owner of the DL model to prove to others that the prediction of a data sample is indeed calculated by the model without leaking any information about the model itself.

## Figures and Tables

**Figure 1 sensors-22-08254-f001:**
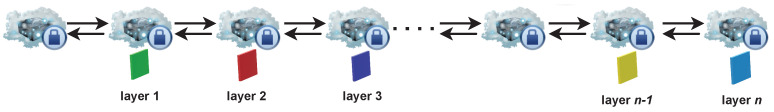
MixNN design overview.

**Figure 2 sensors-22-08254-f002:**
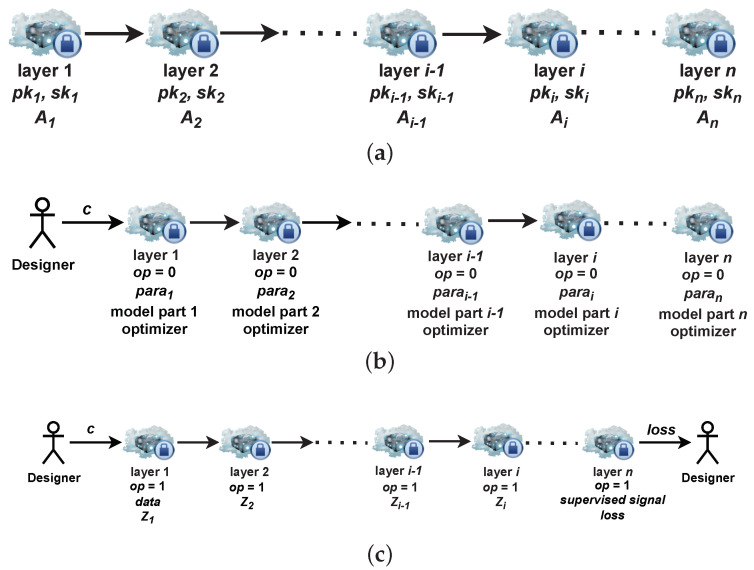
Training phase (*c* is the ciphertext). (**a**) A cascade is constructed. The designer distributes layers on different servers and each layer generates its key pairs and its IP address. (**b**) Model initialization phase. The designer deploys the size of parameters, parts of the model and optimizer on the corresponding layers. (**c**) Forward propagation phase. The designer inputs data into the first layer and the supervised signal into the last layer. Except for the last layer, each layer needs to compute $Z_{i}$. The last layer should compute the loss *l*. (**d**) Backward propagation phase. For layer *i*, layer *i* updates its parameters $W_{i}$ using intermediate gradient $\frac{\partial l}{\partial z_{i}}$ from the previous layer, and prepares $\frac{\partial l}{\partial z_{i-1}}$ for the next layer.

**Figure 3 sensors-22-08254-f003:**
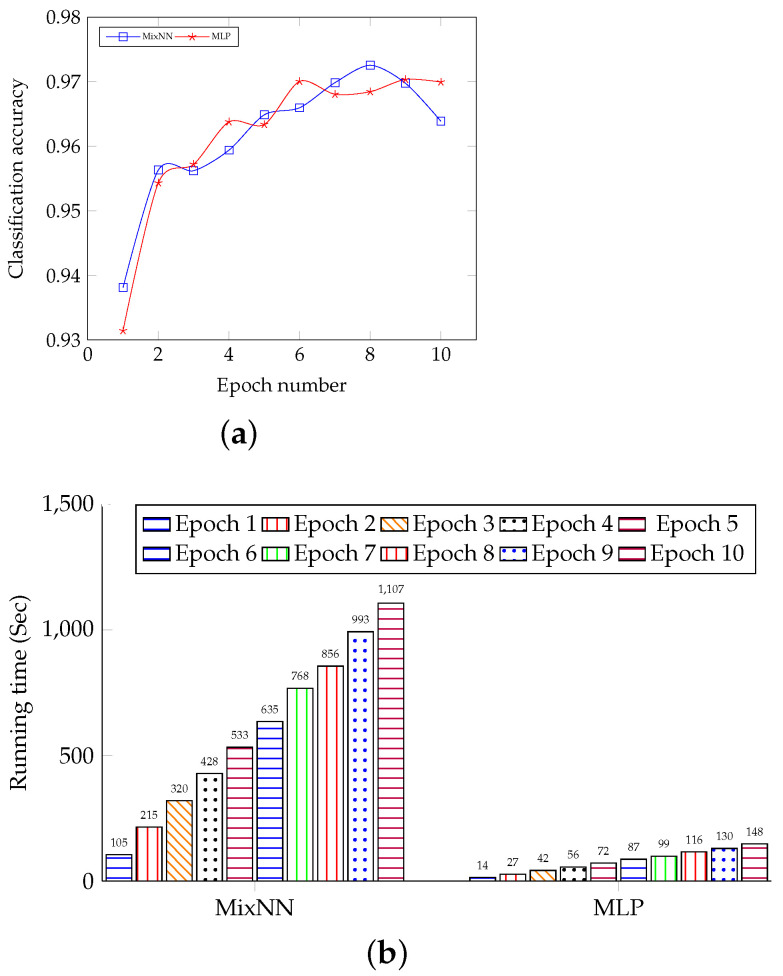
Classification accuracy and running time compared MixNN with MLP. (**a**) Classification accuracy compared Mix-NN with MLP. (**b**) Running time compared MixNN with MLP.

**Figure 4 sensors-22-08254-f004:**
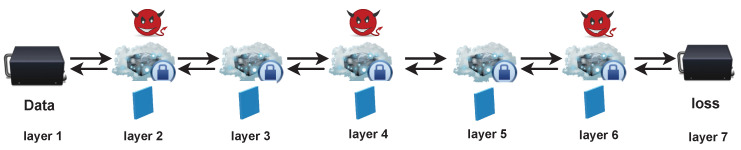
The most severe attack in the second case in MixNN.

**Table 1 sensors-22-08254-t001:** The configuration of MLP in decentralized servers.

Server Index	Operations	Input Dimension	Output Dimension
1	Linear + ReLU	784	128
2	Linear + ReLU	128	64
3	Linear	64	10
4	LogSoftmax	10	10
5	NLLloss	10	1

## Data Availability

Not applicable.

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
