# Peer review of "MixNN: A Design for Protecting Deep Learning Models"

_sensors, 2022, doi:10.3390/s22218254_

Round 1

Reviewer 1 Report

This paper proposed a new approach for protecting deep learning structure and parameters. It hides communication address, layer parameters and operations, and forward as well as backward message flows among non-adjacent layers using the ideas from mixed networks. Yet, the authors may consider the following comments:

1. The Abstract should briefly review the background of the investigation. For instance, why protect deep learning model structure and parameters?

2. In the Introduction section, you should state the scope of the problem investigated. Yet, I think the purpose of writing the paper is not stated briefly and clearly. For instance, in lines 28-29, you said, “As a consequence, model privacy is not preserved.” There are many relevant studies on the safety of deep learning models. Therefore, the stated sentence cannot summarize the literature.

3. Related Work is not comprehensive; many critical works related to reliable deep learning are missing.

4. All formulas need to be numbered. 

5. In the evaluation Section, the paper compared MixNN with MLP. The effectiveness of MixNN is difficult to guarantee, and I suggest offering other DL models for comparison.

6. In the Conclusion section, I suggest pointing out any exceptions and defining unsettled points. In addition, the Introduction has posed one question “As a consequence, model privacy is not preserved.” The Conclusion should indicate what is about the answer. 

Author Response

Dear reviewers and editor,

On behalf of the authors, I’m writing this letter to submit our revised paper “MixNN: A design for protecting deep learning models”, after major revision.

We endeavor to address all the comments and concerns raised by the reviewers. See below the detailed response to each comment. Briefly speaking, we have made the following major changes. The modified parts are in red color in our manuscript.

We have listed some literature to illustrate the challenges to protecting the privacy of the deep learning model.

We have added more related works and discussed them in the related work section.

Efficiency analysis is discussed.

The limitations of this work are presented in the conclusion section.

Chao Liu

T

1. The Abstract should briefly review the background of the investigation. For instance, why protect deep learning model structure and parameters?

Ans: We have added the sentence to describe why we need to protect deep learning model structure and parameters. 

2. In the Introduction section, you should state the scope of the problem investigated. Yet, I think the purpose of writing the paper is not stated briefly and clearly. For instance, in lines 28-29, you said, “As a consequence, model privacy is not preserved.” There are many relevant studies on the safety of deep learning models. Therefore, the stated sentence cannot summarize the literature.

Ans: We agree that this sentence is not accurate and we have modified it. Meanwhile, we have listed some attacks to illustrate the challenges to protecting the privacy of the deep learning model. 

3. Related Work is not comprehensive; many critical works related to reliable deep learning are missing.

Ans: Thanks for your comments. We have added more related works and discussed them in the related work.

4. All formulas need to be numbered. 

Ans: We have modified these.

5. In the evaluation Section, the paper compared MixNN with MLP. The effectiveness of MixNN is difficult to guarantee, and I suggest offering other DL models for comparison.

Ans: We agree that our discussion is not enough. We add a new subsection to discuss it.

6. In the Conclusion section, I suggest pointing out any exceptions and defining unsettled points. In addition, the Introduction has posed one question “As a consequence, model privacy is not preserved.” The Conclusion should indicate what is about the answer. 

Ans: We conclude the limitations of our work in this section. We have modified the sentence which is not accurate. 

Reviewer 2 Report

Manuscript titled "MixNN: A design for protecting deep learning models" proposes a novel design for decentralized neural networks that protects privacy and reduces the likelihood of an attacker gaining control over an entire network and altering output. The proposed NN design opens the door for fully decentralized deep learning and higher security and verifiability of the data that people send to the cloud for processing.

The authors mention the problem of verifiability, multiparty computation, and zero-knowledge to be addressed more in the future.

There are just a few minor issues. The text is written more like a technical report than a scientific paper, but it contains all or almost all the necessary information.

I would suggest providing a public repository for your source code in the paper. This would greatly increase the visibility of the paper and your research.

Author Response

Dear reviewers and editor,

On behalf of the authors, I’m writing this letter to submit our revised paper “MixNN: A design for protecting deep learning models”, after major revision.

We endeavor to address all the comments and concerns raised by the reviewers. See below the detailed response to each comment. Briefly speaking, we have made the following major changes. The modified parts are in red color in our manuscript.

    We have listed some literature to illustrate the challenges to protecting the privacy of the deep learning model.
    We have added more related works and discussed them in the related work section.
    Efficiency analysis is discussed.
    The limitations of this work are presented in the conclusion section.

Please see below in detail how we address each comment.

Chao Liu

I would suggest providing a public repository for your source code in the paper. This would greatly increase the visibility of the paper and your research.

Ans: Thanks for your comments. We will release it as open-source soon.  

Round 2

Reviewer 1 Report

The authors have addressed my concerns.